# Health service responses and help-seeking for women experiencing violence during outbreaks in low- and middle-income settings: A scoping review

Rose Burns*, Manuela Colombini, Neha S. Singh, Janet Seeley

Department of Global Health and Development, London School of Hygiene and Tropical Medicine, London, United Kingdom

* rose.burns@lshtm.ac.uk

## Abstract

During outbreaks women struggle to access essential health services, including services for violence. Services may be disrupted or deprioritised, or women may avoid clinical settings. We conducted a scoping review to understand how health services for violence against women (VAW) were affected in low- and middle-income (LMIC) settings during recent outbreaks, and how women sought help following violence. We reviewed published academic literature reporting on primary research from LMIC settings during recent outbreaks (Ebola, Zika and COVID-19). Four databases were searched: Medline, Embase, Global Health, and Global Index Medicus. Thirty two papers met the inclusion criteria. Data were extracted using a thematic framework focusing on both the supply and demand for services. Experiences during COVID-19 were overrepresented, with no studies identified from other outbreaks. Research spanned 20 countries including a range of services and populations. In the face of lockdowns and reorientation of the health system towards COVID-19, VAW services were restricted or closed despite being essential. Many settings reported shifting services online or to telehealth platforms, raising concerns about digital access and safety, particularly when women accessed services from spaces shared with a violent partner. Some other adaptations included the use of community volunteers and the provision of cash assistance for survivors. Help-seeking varied, with some healthcare settings reporting increases and others decreases in the number of survivors presenting, likely reflecting fluctuating restrictions. Women experiencing violence often sought help from informal sources such as community leaders and family. Help-seeking was further constrained by the economic crisis accompanying COVID-19, including food insecurity and transportation challenges. To prepare for future outbreaks, research is needed to identify which services can be safely and equitably delivered online, and which require in-person provision, as well as to understand a broader range of emerging practices for adapting services to physical distancing, movement restrictions, and economic stress.

**Data availability statement:** All relevant data are within the paper.

**Funding:** The authors received no specific funding for this work.

## Background

Violence against women (VAW) is a global human rights violation and a public health concern that requires a multisectoral response, including from the health system [1]. It is linked to multiple adverse physical, mental, and sexual and reproductive health (SRH) consequences [1]. The connection between VAW and outbreaks has gained attention, particularly during the COVID-19 pandemic, which was accompanied by a 'shadow' pandemic of VAW [2,3]. During this period alone, 31% of women worldwide experienced intimate partner violence (IPV), with the highest prevalence in regions in the Global South [4]. Recent outbreaks have consistently exacerbated violence due to the accompanying economic stress and lockdowns, which trapped women at home with abusive partners [5,6].

During outbreaks like COVID-19 and also Ebola, essential health services for women, including those for maternal, and SRH, were deprioritised or disrupted in many settings [6–8]. Violence against women services such as psychosocial support, medical and multi-sectoral referrals (e.g., housing) may also be affected by outbreaks. Public health emergencies may further limit access to services due to movement restrictions, reduced availability of commodities, increased pressure on the health workforce, service closures, and physical distancing measures affecting in-person service delivery [6,7,9–11]. Even outside of outbreaks, health policy and responses to VAW remain a low priority. Whilst 80% of countries globally have multisectoral VAW policies in place, only 34% include VAW response and prevention as a strategic priority, and just 48% have clinical guidelines for the health sector [12]. The economic and security challenges accompanying crises may further exacerbate barriers to help-seeking [5,6,11]. Whilst normative efforts on how to address VAW have accelerated since the COVID-19 pandemic [13,14], significant gaps remain in understanding what has been implemented in low-and middle-income settings (LMIC) as well as how women seek help [15].

Our study aimed firstly to understand how health services for violence against women (VAW) were affected in low- and middle-income (LMIC) settings during recent outbreaks, and secondly to understand women's help-seeking after facing violence during these periods. Violence against women can be described as a range of acts of gender-based violence that results in, or is likely to result in, physical, sexual or mental harm or suffering [16]. In the literature there are several terms that are often used interchangeably to describe violence experienced by women. Gender based violence (GBV) is an umbrella term referring to violence that can affect men, women, children or LGBTQ+ communities. As part of examining violence against women we also included intimate partner violence, which is one of the most prevalent forms of VAW.

Given that health services may treat various types of violence (e.g., usually physical, sexual and psychological) we included services for all types of VAW, including violence perpetrated by partners, as well as non-partners. We conducted a scoping review of the literature to synthesise evidence on both the supply and demand for health services for VAW during outbreaks of emerging/re-emerging pathogens– such as Ebola, Zika and COVID-19–in order to identify disruptions, adaptations, and

gaps in service delivery and help-seeking. To address this objective we asked two sub-questions: i. to what extent health services for VAW were delivered during recent outbreaks, and what challenges and facilitators shaped service provision (supply side); ii. how women sought help for VAW during outbreaks and which sources of support they tried to access (demand side).

## Methods

### Study design

We conducted a scoping review drawing on Arksey and O'Malley's five stage approach: identifying the research question, identifying relevant studies, study selection, charting the data, collating, summarising and reporting the results [17].

### Search strategy and inclusion and criteria

We searched four databases: Medline, Embase, Global Health, and Global Index Medicus, using search terms related to three concepts: i. health services and health care, ii. violence against women, iii. emergent outbreaks such as COVID-19, Ebola and Zika (see supplementary materials for full search strategy). The search was limited to studies conducted in LMICs (as defined by each of the databases at the time of our search) between January 2014 to May 2024. This timeframe captured the largest Ebola outbreak to date in West Africa (2014–2016), the public health emergency in the Americas following the Zika outbreak (2015–2016), and the containment measures during the COVID-19 pandemic (2020–2023). We focussed on large scale outbreaks that disrupted health systems, restricted access to services, and were associated with increased risks of VAW through mobility restrictions, impacts on women's reproductive health and sexuality, and heightened economic stress within households. On this basis we selected Ebola, Zika and COVID-19, as these outbreaks met these criteria and have been shown in prior research to exacerbate the risks of VAW and barriers to accessing care [4,18,19]. We included articles on healthcare, health policy and community health responses, as well as survivors' help-seeking behaviours. Several terms are used interchangeably (and sometimes ambiguously) in the literature to describe violence experienced by women and other populations. 'Gender-based violence' (GBV) or 'sexual and gender-based violence' describe any harmful act directed at a person because of their (socially prescribed) gender, GBV is often (but not exclusively) perpetrated against women. We also included intimate partner violence (IPV) in our search, which is one of the most prevalent forms of VAW.

The inclusion/exclusion criteria are summarised in Table 1:

### Screening

We initially screened potential sources by title and abstract, and then by full text against the inclusion and exclusion criteria [17]. One author (RB) screened all abstracts and potentially eligible full text articles and extracted the data. Double screening by two co-authors (MC and JS) took place selectively at two different stages of the review, specifically for papers where there was some uncertainty over inclusion. At the title and abstract stage, two reviewers (MC and JS) double-screened 32 papers; and 4 papers at the full text screening stage. Regular discussions between the first author and co-authors throughout the screening process ensured consistency in applying inclusion and exclusion criteria and a consensus on final decisions.

### Data extraction

Data were extracted based on the inclusion criteria to cover both supply and demand side topics using Excel led by the first author (RB). Information including author, title, year published, country, setting, study design, methods, target population, study aim/objectives and study period were also extracted from each article

**Table 1. Inclusion/exclusion criteria.**

|  | Inclusion criteria | Exclusion criteria |
|---|---|---|
| **Population(s)** | - Women survivors of VAW, gender based violence (GBV) or sexual and gender based violence (SGBV); members of communities affected by outbreaks; members of organisations providing services for women survivors (e.g., health workers).<br>- Study populations in LMICs. | - Study populations in high income country settings.<br>- Children or male survivors of violence. |
| **Intervention(s)** | - Studies looking at access to, and delivery of, health service interventions for VAW survivors during disease outbreaks (e.g., Ebola, Zika, and COVID-19), including how outbreaks affected health services and survivors' help-seeking.<br>- Healthcare interventions for survivors (e.g., post exposure prophylaxis (PEP), emergency contraception, treatment of injuries, psychological first aid, mental health care, psycho social support, referrals).<br>- Health outreach and awareness programmes related to VAW. | - Papers reporting exclusively on non-health VAW interventions, including justice and legal aid; prevention; economic empowerment and livelihood interventions.<br>- Papers reporting exclusively on maternal and child health, family planning or HIV/AIDS interventions without GBV/VAW components.<br>- Interventions during HIV/AIDS epidemics.<br>- Interventions during epidemics that are not emergent (e.g., non-communicable diseases). |
| **Outcome(s)** | - Studies describing health services (as outlined above) to address VAW during outbreak periods. | |
| **Study designs** | - Any study design (qualitative, quantitative) that reports results from primary data collection.<br>- Peer-reviewed. | - Commentaries, blogs, opinion pieces and other sources not reporting primary data. |
| **Date or language criteria** | - Published in English.<br>- Studies on contemporary and emergent/re-emergent pathogen outbreaks that have occurred at a large scale in LMICs (i.e., Ebola outbreaks, Zika, COVID-19).<br>- Studies published from 2014 (timeframe of the West African Ebola outbreak). | - Studies published prior to 2014. |

## Synthesis

Findings from the papers were extracted initially using the (deductive) themes of 'supply' and 'demand' and subsequently divided inductively into sub-themes using Excel. This analysis was initially based on the themes identified from papers whose primary focus was our research topic, and was then extended to the remaining papers (many of which only included some incidental data on our research topic). This process was iterative and sub-themes were refined throughout the analysis process through regular team meetings (with RB leading the analysis). This thematic synthesis broadly followed Thomas and Harden's [20] approach, adapted to the synthesis of secondary constructs (e.g., the authors' interpretations of barriers/facilitators to service access) from both quantitative and qualitative data. Thomas and Harden's approach includes inductive coding of findings from papers in order to develop 'descriptive' themes, and following this the development of 'analytical' themes where the reviewers go beyond the individual studies to develop new constructs or explanations that address the review question.

On the 'supply' side, we identified three themes related to health service delivery: i. disruptions to VAW health service delivery during COVID-19, ii. adaptations to VAW health service delivery, iii. continuation of challenges delivering VAW services from 'normal' times. On the 'demand' side, we identified three themes related to survivors' help-seeking: i.

increases and decreases in survivors presenting at health services, ii. women's experiences seeking help with informal sources such as community members and family and iii. economic barriers to accessing health services. Since this was a thematic synthesis of secondary constructs, participant quotes are only included to illustrate authors findings where these were presented in the original studies, and as such they are not included for all themes.

## Results

### Search results

The search yielded a total of 4,859 articles which were imported into Rayyan (https://www.rayyan.ai/) and screened for duplicates, then 3,363 articles were screened for eligibility of which 3,307 were excluded and 56 identified for full text screening. After full text screening 32 articles met the inclusion criteria and were included in the final synthesis (see Fig 1).

### Characteristics of studies

All thirty two articles included focussed on experiences during the COVID-19 pandemic, no papers were found examining experiences during other outbreaks such as Zika or Ebola. Papers presented both data on VAW help-seeking and on health service delivery for VAW during COVID-19. Studies were published between 2021–2024 with no articles appearing from earlier periods. Studies presented primary research conducted in 20 different countries: India [21–24], Nepal [25,26], Pakistan [27], Bangladesh [27–29], Malaysia [30], Romania [31], the Occupied Palestinian Territory [25], Iraq [32,33], Lebanon [34], Kenya [27,35–41], Mali [25], South Africa [37,42,43], Nigeria [27,37,44–46], Ethiopia [47], Burkina Faso [38], Uganda [36,37], Tanzania [36], Brazil [33,48–50], Guatemala [33,50], and Bolivia [25]. Four articles specifically included or

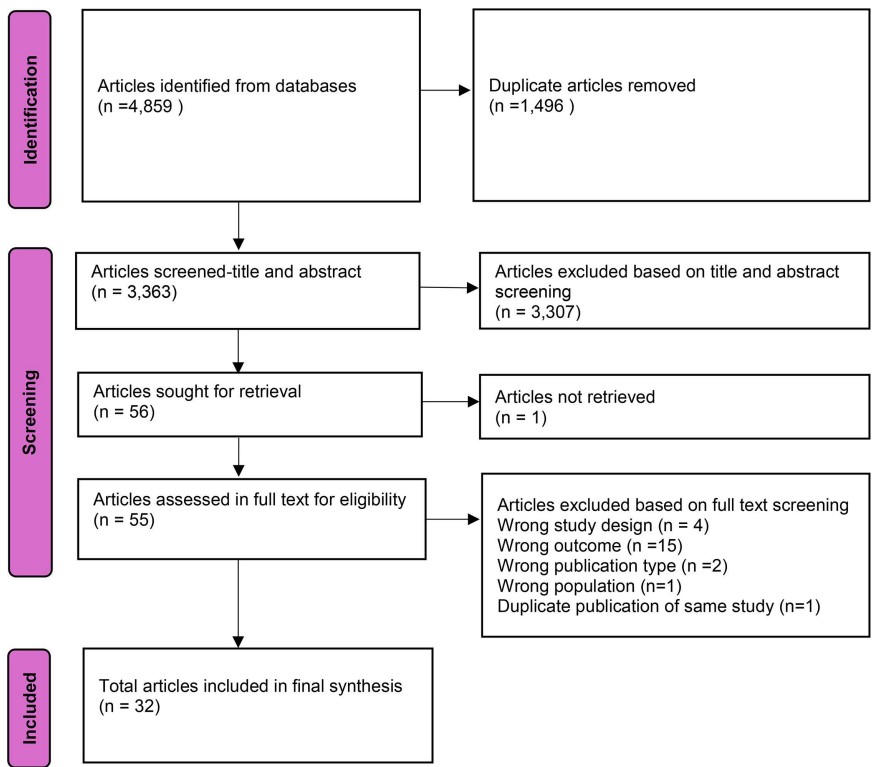

**Fig 1. PRISMA diagram.**

focussed on informal settlements [24,27,33,50], and five on humanitarian settings (e.g., refugee camps or conflict settings) [25,29,32,34,51].

Most studies occurred in a context where VAW, especially IPV, increased (sometimes dramatically) during the pandemic, fuelled by lockdown measures. Many studies had data collection take place during the pandemic however it was difficult to disentangle what challenges were specific to the pandemic, compared to pre/post-pandemic. Papers often did not describe locally specific lockdown details. Given the significant variation in these measures across countries and time periods, it was not always clear how different containment measures impacted service delivery or survivors' help-seeking, especially in retrospective studies.

Articles reported a range of methods, with more than half using qualitative methods. Most studies focussed on violence against cisgender adult women, though some also included child, adolescent or male survivors (with these findings excluded from the review where possible). Many of the studies did not disaggregate the data by age or sex, making it difficult to determine whether the findings were about men, women or children. Studies looked at different types of violence, or different terminology, including VAW, GBV, sexual and gender-based violence (SGBV), domestic violence, IPV, and sexual violence. Eleven studies explicitly focussed on domestic violence or IPV, which was particularly exacerbated during the pandemic. Studies examined a range of health service delivery settings treating survivors, including both specialised VAW services and other broader healthcare services. Sites included non-governmental organisations (NGO) and government health centres, GBV specific services (e.g., 'one stop shops'), outpatient departments, community health settings, family health centres, abortion services, counselling and mental health services, and SRH service settings.

Twelve studies presented data on health service delivery (supply topic), nine on help-seeking (demand topic) and eleven had data on both. First authors from LMIC institutions where the research was conducted appeared to be well represented. Most studies provided only minimal findings directly relevant to our research topic. Instead, they primarily focussed on issues such as increased rates of VAW during the pandemic or the impact of the pandemic on mental health or/and SRH services. While these studies included some incidental data on VAW health service access, only five articles had this as their primary focus (these are marked as 'key papers' in Table 2) [29,33,34,37,47].

We did not conduct a quality assessment of the included studies, in line with scoping review methodology, which focuses on mapping the breadth of the literature rather than assessing the quality of individual study findings.

A description of the 32 articles included in the final review is provided in Table 2:

## Results by theme

This section organises the literature thematically into both our supply and demand side topics. All themes describe experiences during the COVID-19 pandemic as we did not find papers that focussed on other outbreaks such as Ebola or Zika. First, we present the key findings in the literature related to supply themes, focusing on health service delivery for VAW during COVID-19. Then, we present the key findings in the literature related to demand themes, highlighting women's help-seeking during the pandemic.

### 1. Health service delivery for VAW during COVID-19 (supply themes)

**1.a. Disruptions to VAW health service delivery during COVID-19.** Across settings, in-person services were disrupted or closed down during the COVID-19 pandemic due to the containment measures and the reorientations of the health system towards the COVID-19 response. Because many papers lacked detailed descriptions of locally specific lockdown measures, which varied significantly across time and setting, it is difficult to determine which factors were most disruptive. In some settings, essential VAW services were severely impacted. In Bangladesh, in Rohingya refugee camps during the pandemic, SGBV services were initially designated as 'non-essential' and drastically reduced their staff. Participants noted that this de-prioritisation slowed service provision for SGBV [29]:

**Table 2. Summary table with study characteristics.**

| Citation | Study design | Study aim/ objectives | Methods | Target population | Location/ setting | Study period | Findings on health service delivery | Findings on help-seeking |
|---|---|---|---|---|---|---|---|---|
| (Adelekan et al., 2024) | Qualitative | To explore women and adolescent girls' access to sexual and reproductive health services during the COVID-19 lockdown. | 10 focus group discussion sessions held with married adolescents, unmarried adolescents, and older women of reproductive age. | Women and adolescent girls. | Nigeria | Post lockdowns (2021). | Participants saw an increase in the number of women who experienced domestic violence in their communities and no effective response to their needs by the available SRH services. | Harassment by uniformed men made it difficult to access facilities. |
| (Ahmed et al., 2020) | Qualitative | To understand stakeholder perspectives and experiences of healthcare access for non-COVID-19 conditions in informal settlements. | Individual and group discussions with stakeholders, including with 860 community leaders, residents, health workers and local authority representatives. | Stakeholders in informal settlements. | Bangladesh, Kenya, Nigeria and Pakistan (informal settlements). | Pre COVID-19 and during lockdowns. | Only in Kenya were GBV services existent for slum dwellers before the pandemic, there were no reported new GBV services with the onset of COVID-19 lockdowns. | N/A |
| (Alexandru et al., 2021) | Qualitative | To identify measures taken by authorities to improve domestic violence services. | Interviews with 76 specialists who work with survivors of domestic violence. | Domestic violence service providers. | Romania | Post lockdowns (2020–2021). | Specialised GBV services did not allow survivors who tested positive for COVID-19 to access services immediately. | N/A |
| (Barasa et al., 2021) | Mixed methods | An analysis of the indirect health effects of the COVID-19 pandemic in Kenya. | Analysis of secondary quantitative data obtained from the Kenya Health Information System database and a qualitative inquiry involving key informant interviews (n = 12) and document reviews. | Health service users, providers and key informants. | Kenya | Pre, during and post lockdowns. | N/A | Numbers of survivors of sexual violence presenting at an outpatient department (OPD) increased during the lockdown. |
| (Bhatt et al., 2023) | Survey | To assess the prevalence of IPV among married women of reproductive age in Nepal during the pandemic. | An online survey was conducted with 420 participants using a validated questionnaire. | Married women of reproductive age. | Nepal | Post lockdown. | N/A | Low reporting rates and help-seeking behaviour for IPV survivors during COVID-19, most women who sought help approached their family and friends. |
| (Chime et al., 2022) | Cross-sectional | To determine the prevalence and patterns of GBV among victims presenting in a tertiary health facility. | Secondary data from patient records of 710 survivors. | Survivors receiving care. | Nigeria | Comparative, before and after COVID-19. | N/A | At a tertiary health facility there was decreased help-seeking for both sexual and physical/ emotional violence during the pandemic. |

*(Continued)*

| Citation | Study design | Study aim/ objectives | Methods | Target population | Location/ setting | Study period | Findings on health service delivery | Findings on help-seeking |
|---|---|---|---|---|---|---|---|---|
| (Chowd-hury et al., 2022)* | Qualitative | To understand how the COVID-19 pandemic affected SGBV and the provision of services for Rohingya survivors. | Interviews with 13 health or service provider professionals. | Service providers. | Rohingya refugee camps in Bangla-desh (refu-gee camp settings). | Post lockdowns (2020). | Access to the refugee camps, initial designation of sexual and GBV related services as non-essential, com-munications and tele-health, difficulty main-taining confidentiality, and donor pressure affected health work-ers ability to provide essential care and services during COVID-19. Emerg-ing best practices were also reported, including engaging Rohingya volunteers to continue services and adapting pro-gramming modalities and content to the COVID-19 context. | Despite the suspected increase in violence, health care workers noted that the pan-demic lead to a further decrease in service utilisation. Respon-dents described that Rohingya refugees were not legally allowed to own SIM cards due to Govern-ment restrictions. |
| (Decker et al., 2022) | Cross sectional survey | To measure the prevalence of IPV and non-partner household abuse through the COVID-19 pandemic. | A cross sectional survey conducted annually and nationally. | Women aged 15–49. | Burkina Faso and Kenya. | Post lockdown. | N/A | Low help-seeking. Women mostly sought help from family and friends for IPV and household member violence. |
| (Huq et al., 2021) | Qualitative | To explore the experiences of survivors of domestic violence in Mumbai infor-mal settlements. | Interviews with 586 women. | Residents of urban informal settle-ments who registered with a domestic violence support organisa-tion. | India (including informal settle-ments). | During lockdowns. | Counselling services went online, safety issues with speak-ing online were mentioned. | Restrictions meant women survivors were unable to draw on emotional support outside the home from neighbours, friends or family members. |
| (Islam et al., 2023) | Mixed methods | To describe the design and implementation of remote antenatal and postnatal care telemedicine services during COVID-19. | Document review, eighteen interviews and two focus groups with mid-wives, managers, and service users. Quantitative anal-ysis to compare maternity care service use trends before and after implementation. | Midwives, manag-ers and antenatal/ postnatal service users. | Bangla-desh. | Before and after lockdowns. | GBV screening in Bangladesh was integrated with tele-medicine for ante-natal and postnatal care. Because of the economic impacts of the pandemic, cash assistance was integrated with GBV case management. | N/A |

*(Continued)*

| Citation | Study design | Study aim/ objectives | Methods | Target population | Location/ setting | Study period | Findings on health service delivery | Findings on help-seeking |
|---|---|---|---|---|---|---|---|---|
| (John et al., 2023)* | Qualitative | To examine the impact of COVID-19 policies on the availability of GBV prevention and response services in four countries. | Interviews with 80 stakeholders representing different GBV services. | Stakeholders representing different GBV services. | South Africa, Kenya, Uganda, and Nigeria. | During lockdowns (2020). | In all four countries, the government's failure to exempt the provision of multi-sectoral GBV services from initial lockdown restrictions led to confusion, disruption and de-prioritisation of clinical and psycho-social management of GBV. Adaptations including digitalisation and using community health volunteers, and also galvanised efforts to provide coordinated services and care. | N/A |
| (Kaswa, 2021) | A records review | The impact of COVID-19 on access to health services for survivors of sexual assault. | Review of patient records at a 'one stop' sexual assault centre. | Survivors using a sexual assault centre. | South Africa | Before, during and post lockdowns. | N/A | There was a major drop in sexual violence cases presenting at a one stop shop during the 2020 year. |
| (Ke et al., 2024) | Qualitative | To explore the effect of the COVID-19 pandemic on the psychological well-being, perception of risk factors and coping strategies of women and older adults from low-income households. | Telephone interviews with 30 women and 30 older adults. | Women and older adults from low-income households. | Malaysia | During lockdowns (2020). | N/A | Due to the movement restriction measures, participants were unable to seek refuge away from family, as they would have done before the pandemic. Self-coping strategies employed by women included leisure activities and hobbies, along with religious or spiritual practices (e.g., meditation), and reliance on social support. |
| (Khan et al., 2024) | Cross sectional | To assess the perception and actions of oral health professionals towards a rise in domestic violence during the COVID-19 pandemic. | Questionnaire with 200 oral health professionals. | Oral health professionals. | India | Post lockdowns (2021). | Most common barrier faced by oral health professionals managing domestic violence was the lack of training in identifying domestic violence (41.4%). | N/A |
| (Kiarie et al., 2022) | Retrospective time-series | To examine changes in 17 indicators of essential health services across four periods. | Health information system data collection at country-level from health facilities across the country. | People using health facilities. | Kenya | Before and after COVID-19. | N/A | Survivors presenting for sexual violence at health care settings increased. |

*(Continued)*

| Citation | Study design | Study aim/ objectives | Methods | Target population | Location/ setting | Study period | Findings on health service delivery | Findings on help-seeking |
|---|---|---|---|---|---|---|---|---|
| (Lobanov-Rostovsky and Kiss, 2022) | Mixed methods | How the global evidence on psychosocial interventions for female survivors of conflict-related sexual violence applies to the context of the female Yazidi population. | A realist review and eight semi-structured interviews with stakeholders who deliver interventions to female Yazidis. | Stakeholders who deliver interventions. | Iraq (internally displaced people (IDP) settings). | Post lockdowns. | Closure of safe spaces, and online delivery of interventions. Service challenges during COVID-19, including limited privacy to speak over the phone at home, reduced size of group activities like sewing, redirected human resources to COVID-19, which meant that gender matching of staff was not prioritised. | N/A |
| (Mahamid et al., 2022) | Qualitative | To explore mental health professionals' perceptions and concerns on GBV among Palestinian women during the COVID-19 pandemic. | Semi structured interviews with 30 mental health professionals. | Mental health professionals. | Palestine (conflict setting). | Post lockdowns (2021). | Mental health professionals reported different interventions used to help women who faced GBV during the pandemic. Most of these interventions focused on increasing awareness and reducing the harmful effects of GBV. | Several respondents mentioned different cognitive and behavioural coping strategies that women used to deal with GBV during the pandemic including silence, fear of reporting violence, using abuse against children, and only in a few cases, divorce. |
| (Mahapatro et al., 2021) | Qualitative | To analyse the role of social support in the lives of women survivors of domestic violence. | Interviews with 36 women who used the service Mahila Salah and Suraksha Kendra (MSSK). | Women survivors of domestic violence. | India | Unknown, explores lockdown period. | Challenges delivering services online/ over the phone from providers included limited time, the possibility of breach of privacy as the husband was constantly at home, access and private use of phone were not easy. Staff did not have access to patient files during the movement restrictions as they were paper based and kept in the office. | Decline in social support for survivors as they are inside the home with limited contact with their family during the lockdown. |
| (Mahlangu et al., 2022) | Qualitative | To understand men and women's experiences of the COVID-19 national lockdown and its impact and link to women and children's experiences of domestic violence in Gauteng province. | In-depth telephonic interviews were conducted with 37 men and women. | Men and women. | South Africa | During lockdowns. | Closure of services and isolation from social networks during lockdown. | Several participants got support from family members including elders. |

*(Continued)*

**Table 2.** (Continued)

| Citation | Study design | Study aim/ objectives | Methods | Target population | Location/ setting | Study period | Findings on health service delivery | Findings on help-seeking |
|---|---|---|---|---|---|---|---|---|
| (Morais Ranzani et al., 2023) | Cross-sectional | To identify the sociodemographic profile and the characteristics of interpersonal violence against older adults during the first year of the COVID-19 pandemic. | Notification data from epidemiological surveillance, hospitals, out-patient services and other public services. | Older adult survivors of violence. | Brazil | During and after lockdowns. | Low referrals of older adult survivors to security and protection agencies. | N/A |
| (Odorcik et al., 2021) | Qualitative | To analyse health professionals' approach to the identification of VAW and their perception of cases during the Covid-19 pandemic in Family Health Centres. | Semi-structured interviews with 23 health professionals. | Health professionals. | Brazil | Before, during and after lockdowns. | Primary health care providers lacked training, confidence and understanding of referral network when seeing survivors of violence. | N/A |
| (Ochieng et al., 2022) | Modelling | To model the unintended consequences of COVID-19 mitigation measures on sexual violence trends in Kenya. | Secondary data analysis from the Kenya Health Information System. | Patients who received clinical care nationally. | Kenya | Before and after lockdowns (comparative). | The proportion of rape survivors receiving the minimum package of standard care declined during COVID-19. | N/A |
| (Okunola et al., 2021) | Descriptive | To describe survivors (and perpetrators) who received medical care at a sexual assault centre during COVID-19. | Patient data from seventy-four survivors who accessed medical services. | Survivors who accessed medical services. | Nigeria | During and post lockdowns. | Few survivors of sexual violence were seen within 24 hours of the event and only around a quarter had follow-up visits. | N/A |
| (Ottosson et al., 2022) | Mixed methods (implementation science) | To outline iDARE programme implementation in Uganda, Kenya, and Tanzania during the COVID-19 pandemic. | Programme documentation. | iDARE participants and implementers. | Uganda, Kenya, and Tanzania, GBV findings are from Kenya. | Post lockdowns. | Improved identification, management, and response for GBV survivors in United States Agency for International Development (USAID) funded health facilities. | N/A |

*(Continued)*

| Citation | Study design | Study aim/ objectives | Methods | Target population | Location/ setting | Study period | Findings on health service delivery | Findings on help-seeking |
|---|---|---|---|---|---|---|---|---|
| (Raftery et al., 2023)* | Mixed methods | To explore challenges for GBV coordination and service delivery during multiple crises. | 29 remote in-depth interviews with GBV and humanitarian stakeholders, reviewed key policy documents and observed seven GBV task force meetings. | GBV and humanitarian stakeholders. | Lebanon (including humanitarian settings). | Post lockdowns. | Services moved online including group psychosocial support which was provided via WhatsApp, and the GBV taskforce implemented guidance and tools to adapt to remote modalities for GBV services (e.g., training on remote case management. The pandemic overlapped with multiple crises including the 2020 Beirut explosion. High rates of staff burnout and staff safety concerns were reported. | Some service users found remote modalities easier or more anonymous. |
| (Reed et al., 2023) | Qualitative | How COVID-19 affected adolescent girls and young women's (AGYW) experiences of GBV, access to care services, economic and social outcomes, and opportunities for interventions. | 5 Focus group discussions with AGYW. | AGYW affected by GBV. | Kenya | Post lockdowns (2021). | COVID-19 disrupted referrals to violence-related services, and reduced access to both medical services and psychosocial services, reduced community-based mental health and support services, closure of 'safe spaces'. | Preference for seeking help with informal sources such as family members, village elders, and chiefs. |
| (Sharma and Khokhar, 2022) | Cross sectional | To find out domestic violence prevalence and coping strategies among married adults during the COVID-19 lockdown. | Online survey of 94 participants. | Married adults. | India | During lockdowns. | N/A | Participants who did not report it to any agency, often reported they did not feel the need to report, others believed that no action will be taken due to lockdown or anticipated improvement in their situation once the lockdown was lifted. |
| (Sorhaindo et al., 2023) | Qualitative | To examine service adaptations implemented in Bolivia, Mali, Nepal, and the occupied Palestinian territory during COVID-19. | Group and individual interviews among 16 service providers, facility managers and representatives from supporting organisations. | Service providers and key informants. | Bolivia, Mali, Nepal, and the occupied Palestinian territories (multiple settings, including humanitarian). | Post lockdowns (2021–2022). | In Bolivia, the use of a messaging application increased access to confidential GBV support and comprehensive abortion care in cases of sexual violence. | N/A |

*(Continued)*

**Table 2.** (Continued)

| Citation | Study design | Study aim/ objectives | Methods | Target population | Location/ setting | Study period | Findings on health service delivery | Findings on help-seeking |
|---|---|---|---|---|---|---|---|---|
| (Vahedi et al., 2022) | Qualitative | To explore how the Guatemalan GBV prevention and response system operated during the COVID-19 pandemic. | Interviews with 18 key informants. | Key informants working within the GBV prevention and response system. | Guatemala | Post lockdowns. | N/A | GBV service providers reported that survivors' help-seeking was affected by food insecurity and economic challenges. |
| (Vahedi et al., 2023) | Qualitative | To investigate how the COVID-19 pandemic amplified structural violence and GBV in Brazil. | Key informant interviews (KII) conducted with 12 service providers working in sectors related GBV prevention and response. | Service providers. | Brazil (including informal settlements). | Post lockdowns. | GBV service providers prioritised food provision over GBV services in favelas. | Poverty and hunger affected help-seeking. |
| (Vahedi et al., 2024)* | Qualitative | How GBV service providers navigated the process of digitalizing GBV prevention/ response during the COVID-19 crisis. | Key informant interviews (KII) with 51 GBV service providers. | GBV service providers. | Brazil, Guatemala, Iraq, and Italy (multiple settings including informal settlements). | Post lockdowns (2021). | Digitalisation of specialized legal, medical, mental health, and psychosocial GBV care. However, using information and communication technologies (ICT) to offer continuity of specialized care to survivors was not always perceived as comparable to in-person care with safety the most pressing concern. | Lack of digital skills prevented survivors' use of ICT to engage with the GBV response system. |
| (Yirgu et al., 2023)* | Qualitative | To understand the needs and unmet needs of IPV survivors in Ethiopia amid the COVID-19 pandemic. | In-depth qualitative interviews with 24 women who experienced IPV during recent pregnancy. | Recently pregnant IPV survivors. | Ethiopia | Post lockdowns. | Although formal IPV services remained open throughout the pandemic, restrictions resulted in the perception that services were not available which discouraged survivors from seeking help. Pre-existing access difficulties were exacerbated by the COVID-19 pandemic. | Overall, informal sources from family members, neighbours, and village elders appeared to be the first resources women turned to for IPV support. |

\* Key papers.

*"The government... [is] giving more priority to both food and health as life- saving. But, that SGBV is also a part, and that it can also be lifesaving, on that we noticed a little indifference on the part of the government….If they would have given priority to the GBV part as a lifesaver, we could have made our work faster."* [GBV service provider] [29]

Containment measures created multiple additional challenges that affected health workers' ability to provide essential care and services to Rohingya survivors. These included restrictions on access to the camps; physical distancing requirements, which reduced the number of people allowed inside service facilities; constraints on the types of services that could be provided and how they were delivered (given three feet of distance had to be maintained); and donor expectations that service delivery continue despite government restrictions [29]. Similar disruptions were reported in non-humanitarian settings. In Kenya 'Safe spaces' were closed during the pandemic, cutting off access to many VAW related services such as information, social support, and 'mentors' who could provide counselling [41]:

*"you will find us missing our ways because there is no one to advise us."* [adolescent girl or young woman participant, Kisumu, Kenya] [41]

In other sites, services remained open but were overwhelmed by the demands of the pandemic, leading to inadequate care for VAW survivors. In Lebanon a qualitative study with GBV and humanitarian key informants reported high rates of staff burnout, which made it difficult to sustain GBV service provision. Staff safety concerns were also high with threats and theft reported by most service providers [34]. Similarly, in Kenya, Uganda, South Africa and Nigeria, GBV service providers reported that government decisions to reassign health workers to the COVID-19 response reduced the number of providers available to serve the specific medical needs of survivors, including medical forensic exams and psychosocial services [37]. As a result, survivors who did reach health facilities were sometimes told to return at a later date for care [37].

Studies also reported declines in both the availability and quality of services for survivors. A study using data from the Kenya Health Information System documented a decline in the proportion of rape survivors (across all ages and genders) receiving the national minimum package of standard care during the pandemic [35]. For example, the percentage of reported rape survivors receiving recommended HIV post-exposure prophylaxis (PEP) decreased from 61% to 51% [35]. In a study on psychosocial interventions with displaced women survivors in Iraq, it was reported that human resources were redirected to the COVID-19 response, meaning that gender matching of staff was no longer prioritised, which limited women survivors access to female providers [32]. In other settings, such as Romania, GBV service providers within NGO and government settings (working across both health and non-health sectors) reported that specialised services would not immediately accept survivors who tested positive for COVID-19 [31]. In Nigeria adolescents and women of reproductive age reported that although domestic violence increased in their communities during lockdowns, available SRH services did not provide an effective response to their needs [46].

Despite these widespread disruptions, there was also an account of VAW interventions delivered successfully despite the pandemic, although this may have required substantial additional resources [36]. A USAID funded project in Kenya that focussed on locally led solutions was reported by the authors to have improved the identification, management, and response for GBV survivors by a monthly average of 642% over a 10 month period [36]. These improvements were achieved despite a health worker strike [36].

**1.b. Adaptations to VAW health service delivery during COVID-19.**

***Moving VAW health service online:*** Lockdown restrictions on gatherings and social contact substantially curtailed in-person service across health care settings. In response, many VAW services, especially counselling, adapted by rapidly transitioning to remote delivery through online platforms or mobile phones. This shift was discussed in nine of the included articles [23–25,28,29,32–34,37]. However, these adaptations raised important ethical and equity concerns, particularly

around survivor safety and the gender digital divide, which reflects the disparities between men and women in access to and benefits from digital technology.

In Kenyan and South African study sites, authors reported that counselling services were provided remotely to survivors during the lockdowns [37]. Similarly, in study sites in Brazil, Guatemala and Iraq, some legal, medical, mental health, and psychosocial care for survivors was digitalised, including remote legal counselling and online group information or counselling sessions for survivors [33]. Despite these efforts, remote modalities were not always perceived as comparable to in-person care. Providers reported that implicit signs of abuse were difficult to spot or document when a survivor is seen online [33]. Safety concerns were frequently highlighted as a key limitation of remote service delivery with several sites noting the risk if perpetrators could overhear or observe survivors reporting violence or participating in safety planning via the phone or online platforms. In an Indian site, where counselling services moved online, women were hesitant to discuss abuse if their husband could overhear them, fearing potential repercussions [24]. Similar concerns about privacy were reported among internally displaced women survivors in Iraq who were engaging with psychosocial interventions online or over the phone [32]. In Bangladesh, women risked disclosing their experiences of SGBV to others when engaging with case workers, volunteers, and other service providers over the phone in public or in their homes [29].

Limited digital access also posed significant barriers tor safe, effective and equitable service provision. In Bangladesh, GBV screening was integrated into telemedicine services for antenatal and postnatal care [28]). However both survivors and providers often lacked reliable access to internet or telephones, and phones were often owned or controlled by husbands or other family members [28]. Similarly in Brazil, Guatemala and Iraq, barriers included limited access to reliable information and communication technology (such as adequate battery life or headphones), insufficient digital skill (for example connecting to Wi-Fi or sending messages) and the cost of mobile data [33].

These challenges were particularly acute in humanitarian settings and among marginalised populations. In Bangladesh, Rohingya refugees, including Rohingya staff/volunteers living in camps, were not legally allowed to own SIM cards due to government telecommunication restrictions implemented before the pandemic [29]. These restrictions significantly constrained communication between health workers and survivors, particularly affecting counselling, outreach activities, referrals, and other essential services. Similarly, in Nigeria, Kenya, South Africa and Uganda, providers reported that structurally excluded populations, such as adolescent girls and young women, often lack confidential access to phones or other digital technologies needed for online counselling [37]. In some cases, digital services were used to triage survivors who needed in-person services. For instance, in a study on GBV coordination in Lebanon, group psychosocial support was provided via WhatsApp using a mixed approach of chats, voice messages and live calls [34]. However, high-risk cases were prioritised for in-person services [34].

Despite these limitations, several studies also reported perceived advantages of remote modalities. In Lebanon some service users preferred remote modalities because they reduced difficulties with transportation and infection risks, and provided greater anonymity for LGBTQ+users [34]. Some providers felt women might be more comfortable to disclose violence over the phone than in face to face settings [28]. In some settings, virtual counselling was conducted in groups of four to six women survivors, and group participation was seen to improve communication and learning between women [33]. Digital technologies were also used for broader awareness-raising and referral initiatives. In Bolivia a social media campaign focused on educating people about their legal right to medical abortion in cases of sexual violence, and the use of a messaging application confidentially linked survivors with providers for medical abortion [25].

However not all services were able to transition to remote delivery. At an Indian NGO, counsellors reported that they could not offer services until the lockdown eased because survivors' phone numbers and records were stored in paper-based files kept at the office, which providers were unable to access due to movement restrictions [23].

***In-person VAW health service delivery adaptations:*** Alongside the shift towards remote services, several studies described adaptations to maintain in-person service delivery during the pandemic. These adaptations included new guidance, strengthened partnerships, and the use of volunteers and economic interventions. In Lebanon, the pandemic

PLOS Global Public Health

overlapped with multiple crises including the 2020 Beirut explosion compounding challenges for providers. In response, the (national) GBV taskforce implemented harmonised guidance and tools to support field-level actors in adapting to remote GBV service delivery, including training on remote case management [34]. In South Africa, service providers reported that the shift to online meetings strengthened collaborations across organisations, as smaller local organisations were able to connect more easily with larger NGOs and private foundations [37]. In Kenya, given the challenges with digitising, some organisations experimented with using Community Health Volunteers (CHVs) who were trained to provide psychological first aid in-person during the lockdown and helped refer survivors to health facilities. The approach was reported as successful in serving some hard to reach women, although some providers expressed concern that percep- tions of CHVs as non-professional counsellors could undermine confidence in services in the longer term [37]. Similarly, in Bangladesh, when professional staff could not access camps, engaging Rohingya volunteers, including psycho-social support volunteers, was described as an emerging best practice [29]. Volunteers became essential to continuing services by visiting households and referring cases [29].

Programme activities were also adapted to address the broader socioeconomic impacts of the pandemic. In Bangla- desh, a programme pivoted to making face masks (with livelihood components) and alongside this activity women expe- riencing violence were referred for services [29]. In other sites adaptations included economic interventions for survivors. Interestingly in Brazil, GBV providers used their limited resources for food provision rather than GBV services because of the enormous economic hardship and structural violence that favela communities were facing [50]. Similarly, in Bangla- desh, the United Nations Population Fund (UNFPA) piloted integration of cash assistance within GBV case management services [28]. These adaptations were most commonly reported in contexts facing extreme economic repercussions from the pandemic.

**1.c.  Continuation of challenges in delivering VAW health services from 'normal' times.**  Several studies described challenges in VAW service delivery that appeared to pre-date the COVID-19 pandemic, suggesting the persistence of longstanding systemic barriers. These included gaps in provider training, weaknesses in referral systems, and limited availability of services in some settings.

Training gaps among healthcare providers were commonly reported. In two Family Health Centres in Brazil, a qualita- tive study showed primary health care providers lacked training to properly carry out identification and referral of survivors, and understand referral networks [49]. Similarly, in a study with oral health professionals in India, whilst over half had screened new patients for visible signs of domestic violence, the most common barrier they encountered in managing survivors was insufficient training in identifying domestic violence (41%) [52]. Weak referral pathways also appeared to be a longstanding challenge. A cross-sectional study in Brazil examining cases of violence against older adult survivors (men and women) showed that for older adults after being seen in hospitals, outpatient services and other public services, referrals to security and protection agencies were relatively low [48]. In Palestine, counsellors reported that some GBV survivors were reluctant to engage with mental health services, partly due to perceptions that that counselling was only for 'mad' people, or a preference for medication rather than counselling [51]. These attitudes are likely to pre-date the pan- demic and reflect broader stigma surrounding mental health care.

In some contexts, services for survivors were already limited or non-existent prior to COVID-19. A study with stake- holders in Bangladesh, Kenya, Nigeria and Pakistan found that GBV services for slum dwellers existed only in Kenya before the pandemic [27]. In Ethiopia, services provided by institutions such as women's affairs and health facilities were reported to be particularly limited at the local administrative (Kebele) level [47].

## 2.  Women's help-seeking after experiencing violence during COVID-19 (demand themes)

**2.a.  Changes in the number of survivors presenting at health services.**  Studies presented a mixed picture regarding whether health facilities experienced increases or decreases in the number of VAW survivors presenting for care, likely reflecting differing lockdown contexts.

In Kenya, one study reported that the number of survivors of sexual violence presenting at an outpatient department (OPD) *increased* at a monthly rate of 0.15 cases after March 2020 during the lockdown [53]. Another Kenyan study analysed national health facility data collected from health facilities and compared different time periods including pre-pandemic, two time periods during the pandemic, and a period during which there was a health worker strike [39]. The number of survivors presenting for sexual violence also *increased* by 8% during the pandemic, even when presentations for other concerns, such as OPD visits and HIV testing, were decreasing [39].

In contrast, a study from Nigeria suggested reduction in service utilisation. At a GBV unit within a tertiary health facility, analysis over a three year period showed *decreased* presentations for both sexual and physical/emotional violence during the pandemic, with cases peaking pre-pandemic in 2019 and declining in 2020 during COVID-19 [45]. Despite reduced help-seeking overall, the proportion of cases of physical/emotional violence (96%) to the total cases of GBV per year was higher in 2020 during the pandemic compared to 2019 and 2018. This may reflect increases in physical and other forms of violence during the lockdown conditions [45]. Similarly, in South Africa, a 'one stop shop' for health service delivery for sexual violence reported a major *decrease* in cases, with only about half (451) of the annual average cases of sexual violence cases presenting in 2020 compared to pre-pandemic years [43].

However, interpreting these trends is challenging. In all four of the studies mentioned above [39,43,45,53], the authors do not specify whether survivors were presenting for IPV or violence perpetrated by non-partners, and the age and gender of survivors is not reported. As a result, it is difficult to determine whether they reflect changes in patterns of violence (e.g., increases/decreases in violence) or if they reflect changes in help-seeking. It is also unclear to what extent children were included in the data, and children may have experienced quite different patterns of violence during stay-at home orders and school closures.

Beyond the number of cases presenting, other indicators of help-seeking behaviour are also important, including delayed presentation and low follow-up rates, which can have significant implications for time-sensitive interventions such as HIV post-exposure prophylaxis (PEP) and emergency contraception. In a descriptive study of survivors of sexual assault who received medical care at a free 'one-stop', sexual assault clinic in Nigeria from mid-2020 to mid-2021 out of 74 women and girls presenting only seven survivors were seen within 24 hours of the event, and around a quarter had follow-up visits [44]. However, because comparable pre-pandemic data were not provided, it is unclear whether these patterns reflect pandemic-related disruptions or longer-standing trends.

In several studies, such as cross sectional studies, it was unclear whether the pandemic influenced changes in help-seeking because pre-pandemic comparisons were not available, and in many settings survivors rarely seek help from formal services regardless of outbreaks. For example, in one study in Nepal, only 14% of the survivor participants (IPV survivors who were women of reproductive age) reported seeking help [26]. Barriers to help-seeking included economic stress, fear of separation from children, fear of retaliation from partners, limited family and community support and social isolation, travel restrictions, and higher distress levels during the pandemic [26]. Similarly, in an Ethiopian site, none of the pregnant IPV survivors in the study sought help at health services during the height of the COVID-19 pandemic [47]. Although many formal IPV services, such as police, courts, and health centres, remained operational, lockdown measures contributed to widespread perceptions that services were unavailable, which discouraged survivors from attempting to access them [47],

**2.b. Turning to community and family networks for support.** Given the challenges in accessing formal health services during COVID-19, and in some settings reduced help-seeking, many women turned to community members or their family for support. This pattern may have been especially pronounced in cases of IPV, influenced by factors that pre-date the pandemic. In many contexts, women seek support from family or community networks in the hope of preserving their marriage and avoiding the social and economic consequences of a divorce. Help-seeking from informal sources was described in seven articles [22,24,26,30,38,42,47], including two previously discussed (section 2a), where survivors rarely sought help from formal sources. We first summarise patterns of help-seeking from community members reported by

study authors, some of which may reflect pre-pandemic dynamics, and then examine how pandemic-related restrictions disrupted these informal support pathways and shaped survivors' coping strategies.

In Nepal, among women of reproductive age who experienced IPV and sought help during the pandemic, the most common sources were family members (43%), followed by friends (38%), neighbours (10%), and others (10%) [26]. In a cross sectional survey in Burkina Faso and Kenya, over half of women did not seek help for the violence they experienced during the pandemic period [38]. Among those who did seek support, help-seeking was concentrated on informal help (32–43%), particularly the woman's own family, the husband/partner's family, or friends [38]. Authors note that these patterns are consistent with pre-pandemic global trends. Qualitative studies similarly highlighted the importance of community-based support mechanisms. In South Africa, a qualitative study showed that for IPV during the lockdown couples used (family) elders who gave advice [42]. Similarly, in a study in Ethiopia with pregnant survivors of IPV, family members, neighbours, and village elders appeared to be the first resources women turned to for IPV support [47]. The majority of women described these support systems as valuable for facilitating negotiations with their partner, as well as providing temporary housing in order to de-escalate violent situations [47].

However, COVID-19-related restrictions also disrupted help-seeking from informal and community sources of support. Physical distancing, stay home orders, and restrictions on movement also hindered women from receiving support for IPV from neighbours, family, friends, and religious leaders compared to pre-pandemic [47].

*"Before coronavirus I used to tell the mosque Imam and village elders when we have disagreements. But after coronavirus, there were movement restriction, and I could not meet anyone." [IPV survivor,* Southern Nations, Nationalities, and Peoples' Region, Ethiopia] [47].

In Malaysia, women and older adults were often unable to leave the household to seek refuge away from family, as they would have before the pandemic. Similarly, in India women found it hard to draw on emotional support outside the home from neighbours, friends or family members during the pandemic [24]. In some cases, survivors simply tried to cope on their own. In India of 7 men and women study participants who faced domestic violence during lockdown, half of the survivors chose to ignore it and around half used yoga or meditation to cope [22]. Participants (men and women) who did not report the violence to any agency, stated they did not feel the need to report, believed that no action will be taken due to the lockdowns, or anticipated that the situation might improve once restrictions were lifted and there was less interaction with their partners [22].

**2.c. Economic barriers to accessing health services.** Across many settings, economic crises accompanied the pandemic and impacted a range of sectors. In two studies in Brazil and Guatemala the pandemic and containment measures meant that survivors, especially marginalised groups such as Black people, trans people, LGBTQ+ people, migrants, sex workers and survivors living in favelas, faced economic stress which affected their help-seeking [50,54]. This included poverty, food insecurity, unemployment, lack of electricity, precarious labour, and weakened transportation infrastructure [50,54]. The authors found that survivors often faced dire food insecurity (as well as water and sanitation needs) and prioritised meeting basic survival needs over seeking support from GBV treatment and care [50]:

*"They are the families in poverty, below the poverty line. They are those families that come, for example, to the place and ask: 'I just want food. I do not care if I get beaten, if my husband drinks every day, and comes drunk and hits everybody. No, I want food.'"[service provider, Brazil]* [50]

Economic dependency also made it more difficult to escape violent partners and seek help, and movement restrictions and price surges limited access to public transportation [54]. A similar picture emerged in African settings. In Ethiopia distance to services was already a barrier during 'normal' times, however this worsened during COVID-19 with increased

transportation costs making it harder for women attempting to use formal IPV services [47]. In a qualitative study in Nigeria, Kenya, South Africa, and Uganda, respondents described how restrictions on public transportation prevented or delayed survivors' access to medical clinics, especially for those living in rural areas [37]:

> *"We also have challenges in terms of physical visits because some survivors are not able to access the centre because of restriction in movement."* [Clinical service provider, Nigeria] [37]

Across regions, economic concerns were heightened during the pandemic especially for marginalised groups or those living in rural areas or informal settlements, with food insecurity and transportation challenges affecting survivors' ability to seek help from health services.

## Discussion

To our knowledge, this is the first review to examine health service delivery and help-seeking for survivors of VAW during recent outbreaks in LMIC settings. Given that outbreaks exacerbate VAW and disrupt health service access for survivors, an evidence synthesis is essential in order to identify opportunities for strengthening service delivery. Although our search focussed on multiple outbreaks (Ebola, Zika, COVID-19) over the decade preceding the search, all 32 papers included only examined experiences during COVID-19. This increase in research interest may have partly been triggered by the widespread public attention to intimate partner violence (initially in high income settings) during the COVID-19 lockdowns, when women were forced to stay at home with violent partners [55]. Our findings indicate that VAW health service delivery was widely disrupted during COVID-19 with a variety of services affected by movement restrictions, staff reallocation to the COVID-19 response, lower service quality, being designated as 'non-essential', and closure of services such as safe spaces. Some of these were challenges that may have pre-dated the pandemic such as inadequate training and referrals. Many articles described how VAW services attempted to adapt during the pandemic. Services, especially counselling, moved online or to telehealth service delivery. However, remote delivery introduced safety challenges for survivors who lacked private spaces in which to communicate, and this posed additional barriers for more marginalised groups (such as populations in humanitarian settings or youth) who may lack digital access. Some services adopted new delivery modalities with volunteers, alongside economic interventions used to respond to survivors' basic needs.

Evidence on help-seeking was mixed. Some clinical settings reported increases in survivors presenting for care during the pandemic, while others reported decreases. These differences may reflect variation in the types and timings of lockdowns which influenced survivors' ability or willingness to use the health system. In many cases survivors relied on informal sources of help like their families or community members, and the economic crisis that accompanied the pandemic globally presented challenges to help-seeking with transportation challenges exacerbated by the movement restrictions.

Despite calls over the last decade to ensure the continuity of essential health service delivery for women during outbreaks, this review shows that across LMIC settings, VAW services were de-prioritised or closed down during COVID-19. The papers included in this review were diverse in terms of study design, regions where research was conducted, experiences of lockdowns, and existing health infrastructure. Papers also differed in the inclusion of both perspectives on health service provision (these papers often drew on data collected from providers or from facilities) and access to health services (these papers often drew on data collected from service users). However, a consistent picture emerged showing there were significant challenges for women to access health care after facing violence. Although COVID-19 generated political attention to this topic, specifically to the impact of stay at home orders when survivors were locked down with their violent partners, it is not clear whether this will translate into long term investment or policy change. A substantial body of research has examined increases in the prevalence of some forms of violence during COVID-19, across different settings [4,56–58]. However, this body of literature provides limited insights into what responses were available to survivors during the pandemic, and what could be learnt from this for future outbreaks. There is also a body of literature looking at the

secondary impacts for COVID-19 on the health system more broadly (e.g., routine immunisation, SRH services) [59,60]. Despite significant focus on these two topics, they have been examined in parallel rather than together: therefore this scoping review provides a timely contribution.

However, many of the papers we included only look at VAW health service delivery as a very small component of their study, therefore further work is needed to build a picture of how LMIC settings might strengthen VAW service delivery during outbreak containment measures. There is a significant research gap in experiences from the range of other (non-COVID-19) outbreaks that have also shattered heath systems and exacerbated violence. This gap is striking considering that outbreak responses to Ebola (in West Africa and Democratic Republic of the Congo) and Zika (in the Americas) have also been critiqued as gender blind and as having neglected essential services for women [11,18,61,62].

The literature included in our review indicates that many LMIC health systems face significant gaps in terms of resilience and readiness to deliver VAW services during outbreaks like COVID-19. The concept of resilience describes how services continued, or were disrupted, adapted or used different resources in order to deliver services [63]. Most of the adaptations reported during COVID-19 involved remote delivery of counselling services online. However, these approaches were often perceived as inferior to in-person services delivery given they were fraught with safety concerns and inequitable digital access. In preparation for future outbreaks, further evidence is needed on digitalisation of VAW services, including what can be done safely online and what poses safety and access challenges. There are also significant gaps in understanding other types of service adaptations. Although some examples were reported, additional research, particularly from outbreaks beyond COVID-19, is needed to understand emerging best practices for delivering services during physical distancing measures, as well as how survivors might manage disruptions to transport, movement restrictions, and pressing economic concerns.

Despite repeated calls to examine outbreaks through a gender lens, we also found, as others have noted [64], that many COVID-19 studies did not present age, gender or sex disaggregated results. This makes it difficult to interpret whether changes in numbers of survivors presenting at health services were reflective of help-seeking patterns or reflective of increases/decreases in specific types of violence. For example, we know that in some settings the risk of violence against children increased during school closures and periods of heightened economic stress inside families [65]. In some papers that we included, it was unclear whether the health service data pertained to children or adults.

### Strengths and limitations

We did not conduct a search of grey literature, which may partly explain why there was no research found on Ebola, Zika or other recent emergent outbreaks. We assume that some research is available in the grey literature such as NGO reports. Similarly, the time period during which we conducted the search was shortly before Mpox was declared a public health emergency of international concern. Given there are important gender considerations around Mpox [66], this scoping review missed any literature emerging on this topic.

We also only included articles published in English which might partly explain why we did not identify any literature on outbreaks such as Zika.

### Conclusions

This scoping review reveals a striking focus on experiences during COVID-19 in the literature. The literature we identified has highlighted that despite a longstanding consensus that life-saving services for survivors of violence must be maintained during outbreaks, such services were deprioritised or closed down in many LMIC settings during the COVID-19 pandemic. Most adaptations reported focused on remote service delivery, which often introduced significant safety challenges. To strengthen VAW service delivery during future outbreaks, further work is needed to determine which service components can be ethically delivered remotely and online. Further research is also needed to explore a broader range of health service delivery good practices, including beyond the COVID-19 pandemic, that might have emerged in response

to movement restrictions, economic pressure, and physical distancing in health facilities as well as how women sought help.

## Supporting information

**S1 Table. Search strategy Medline.**
(DOCX)

**S2 Table. Search strategy Embase.**
(DOCX)

**S3 Table. Search strategy Global Health.**
(DOCX)

**S4 Table. Search strategy Global Index Medicus.**
(DOCX)

**S1 Checklist. PRISMA checklist.**
(DOCX)

## Acknowledgments

Thank you to Kathleen Perris at the London School of Hygiene and Tropical Medicine library for her support in developing the search strategy.

## Author contributions

**Conceptualization:** Rose Burns, Manuela Colombini, Neha S. Singh, Janet Seeley.

**Data curation:** Rose Burns.

**Formal analysis:** Rose Burns, Manuela Colombini, Janet Seeley.

**Investigation:** Rose Burns, Manuela Colombini, Janet Seeley.

**Methodology:** Rose Burns, Manuela Colombini, Neha S. Singh, Janet Seeley.

**Project administration:** Rose Burns.

**Supervision:** Rose Burns, Manuela Colombini, Neha S. Singh, Janet Seeley.

**Visualization:** Rose Burns.

**Writing – original draft:** Rose Burns.

**Writing – review & editing:** Rose Burns, Manuela Colombini, Janet Seeley.

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
