## [Decision Letter · Decision Letter 0]

20 Aug 2025

PGPH-D-25-00671

Health service responses and help-seeking for women experiencing violence during outbreaks in low- and middle-income settings: a scoping review

Dear Dr. Burns,

Thank you for submitting your manuscript to PLOS Global Public Health. After careful consideration, we feel that it has merit but does not fully meet PLOS Global Public Health’s publication criteria as it currently stands. Therefore, we invite you to submit a revised version of the manuscript that addresses the points raised during the review process.

We look forward to receiving your revised manuscript.

Kind regards,

Muriel Mac-Seing, PhD

Academic Editor

Journal Requirements:

2. We have amended your Competing Interest statement to comply with journal style. We kindly ask that you double check the statement and let us know if anything is incorrect.

3. We note that your Data Availability Statement is currently as follows: “All relevant data are within the manuscript.”

Additional Editor Comments (if provided):

Reviewers' comments:

Reviewer's Responses to Questions

**Comments to the Author**

1. Does this manuscript meet PLOS Global Public Health’s publication criteria? Is the manuscript technically sound, and do the data support the conclusions? The manuscript must describe methodologically and ethically rigorous research with conclusions that are appropriately drawn based on the data presented.? Is the manuscript technically sound, and do the data support the conclusions? The manuscript must describe methodologically and ethically rigorous research with conclusions that are appropriately drawn based on the data presented.

Reviewer #1: Yes

Reviewer #2: Partly

Reviewer #3: Yes

2. Has the statistical analysis been performed appropriately and rigorously?

Reviewer #1: N/A

Reviewer #2: N/A

Reviewer #3: N/A

3. Have the authors made all data underlying the findings in their manuscript fully available (please refer to the Data Availability Statement at the start of the manuscript PDF file)?

The PLOS Data policy requires authors to make all data underlying the findings described in their manuscript fully available without restriction, with rare exception. The data should be provided as part of the manuscript or its supporting information, or deposited to a public repository. For example, in addition to summary statistics, the data points behind means, medians and variance measures should be available. If there are restrictions on publicly sharing data—e.g. participant privacy or use of data from a third party—those must be specified.requires authors to make all data underlying the findings described in their manuscript fully available without restriction, with rare exception. The data should be provided as part of the manuscript or its supporting information, or deposited to a public repository. For example, in addition to summary statistics, the data points behind means, medians and variance measures should be available. If there are restrictions on publicly sharing data—e.g. participant privacy or use of data from a third party—those must be specified.

Reviewer #1: Yes

Reviewer #2: Yes

Reviewer #3: Yes

4. Is the manuscript presented in an intelligible fashion and written in standard English?

Reviewer #1: Yes

Reviewer #2: Yes

Reviewer #3: Yes

5. Review Comments to the Author

Reviewer #1: This study shines a light on a reportedly underexplored aspect of the literature the impacts of disease outbreaks on the impact the provision of and access to health services for violence against women. The authors provide a much-needed description on the published primary studies addressing this topic in the English language, succinctly highlighting the major takeaways from the studies included. The paper includes a very helpful breakdown of the key themes in the findings reported by authors in a number of settings in LMICs and serves to highlight the considerations for future research, interventions, and advocacy. Some minor changes are needed to improve the manuscript and prevent any misinterpretations of the findings reported.

Reviewer #2: This scoping review aims to identify opportunities for strengthening violence against women (VAW) health services during outbreaks in low- and middle-income countries (LMICs), using a thematic synthesis approach. The objectives are clearly stated and the search strategy is well constructed. I appreciated the authors’ transparency regarding the limitations of both their review process and the included studies. However, I have significant concerns about key aspects of the methodology and the overall contribution of the article. In particular, the analytical and synthesis processes are not described in sufficient detail to assess the validity or credibility of the findings. Although the authors refer to a “thematic synthesis approach”, the themes presented appear more categorical in nature, raising questions about the depth of the analysis. Without clarity on how themes were developed, the results are not reproducible, and the review lacks the transparency expected in rigorous scoping reviews. Additionally, the discussion remains quite superficial and does not fully address the article’s stated aim, limiting the paper’s added value to the literature. Despite these major concerns, I have provided detailed comments, both major and minor, because I believe the manuscript has potential. With substantial revisions in three key areas (1. Clarifying the analytical and synthesis methodology, 2. Aligning the presentation of results with the stated synthesis approach, and 3. Significantly deepening the discussion in line with the paper’s stated aim), I encourage the authors to consider resubmitting a new version at a later stage.

MAJOR COMMENTS

INTRODUCTION

While the authors define VAW and cite statistics on IPV, it remains unclear whether the review focuses solely on IPV or considers other forms of VAW (e.g., sexual violence, femicide, trafficking, early/forced marriage, etc.) Given the broad use of the term VAW, it would be helpful to clearly define what types of violence are included within the scope of this review (and later on, in the included articles) and justify this focus. This clarification would improve the conceptual clarity and ensure alignment between the stated objectives and included evidence.

METHODOLOGY

Study Design

Line 73: The manuscript refers to the use of Arksey and O’Malley’s scoping review framework, stating that six stages were followed. However, the original framework includes five stages. Could the authors clarify whether an additional stage was added, and if so, what it entailed? It would be helpful to explicitly state any adaptations made to the original framework to enhance methodological transparency.

Line 73: The authors indicate that Stage 1 of Arksey and O’Malley’s framework (identifying the research question) was followed, yet the specific research question guiding the review does not appear to be stated. For clarity and coherence, I recommend explicitly including the research question that informed the scope and direction of the review.

Lines 83-85 : The manuscript states that the review focused on outbreaks “potentially linked to VAW”. It is unclear how this criterion was defined or operationalized during study selection. Could the authors clarify how the link to violence against women was assessed or applied in determining which outbreaks to include or exclude? Greater transparency on this point would strengthen the rationale for the selection of cases.

Screening

Lines 89-93: The description of the screening process lacks clarity and appears inconsistent. The manuscript states that "double screening by multiple authors took place at two different stages of the review," specifically noting that 32 papers were double screened at the title and abstract stage, and 4 at full text. However, this is difficult to reconcile with the reported exclusion of 3,307 records at the title and abstract stage (as seen in PRISMA figure). Could the authors clarify exactly which stages were double screened, how these 32 and 4 papers were selected for double screening, and whether any inter-rater reliability or consistency checks were performed? How were conflicts resolved? A more detailed explanation would improve transparency and rigor.

Data extraction

Lines 97-98: The statement that "data were extracted based on the inclusion criteria to cover both supply and demand side topics using Excel" lacks sufficient detail. It is unclear what specific data were extracted from the included studies, what the extraction process entailed, how it differed for quantitative vs. qualitative studies, and how it aligned with the review objectives. Could the authors clarify what variables or information were extracted (other than identifying study information like titles, years, countries), whether a standardized extraction form was used, and how the data informed the thematic synthesis? Providing this information would enhance transparency and help readers understand the foundation of the analysis.

Synthesis

Line 101: The manuscript states that a thematic synthesis approach was used to analyze the data. However, there is no information provided on the specific steps undertaken in the analysis. As there are multiple approaches to thematic synthesis (especially in a scoping review vs. systematic reviews), it would be helpful for the authors to cite a methodological reference that guided their approach especially as the results appear to be categorical rather than thematic. As the results appear to be descriptive rather than interpretive, a descriptive approach might be more appropriate. Additionally, more detail is needed on what data were analyzed and synthesized (primary constructs? secondary constructs?), how the thematic analysis was conducted, who conducted it, which software was used, and how themes were identified and refined. Similarly, authors need to report the synthesis process (synthesizing qualitative findings with quantitative findings, primary studies’ interpretations, etc.) Also, what quality control mechanisms were used to ensure the credibility of findings? This information is essential for assessing the rigor and reproducibility of the synthesis.

RESULTS

Overall

While direct quotations from primary studies are included in some parts of the results, they are not used consistently across all themes. I encourage the authors to incorporate illustrative quotations throughout the results section to support the thematic analysis (if this was indeed the type of analysis performed, as the manuscript is missing information in the Synthesis section) and enhance the credibility and depth of the findings.

In several instances, the manuscript refers to country-level findings using broad phrases such as “South African providers reported”, “one study in Nepal”, “In Lebanon...”. However, it is unclear (from both text and Table 2) whether these statements reflect national-level experiences or the specific contexts in which the original studies were conducted (e.g., within an NGO, a government program, or a particular region). I recommend clarifying whether the various findings are drawn from individual study contexts rather than presented as representative of the entire country, to avoid overgeneralization. Instead of naming the countries in which the findings were observed, I recommend providing information about the study context, aim, participants, etc. to better contextualize the findings. I feel like the country is less relevant than the specific context in which the study was conducted (refugee camp, favelas, etc.) In other words, instead of focusing on the country, I would advise focusing on the context and reach of the study, like it was done for reference #35 (lines 197-201 “In a descriptive study […]”).

Theme 2a Patterns: The theme titled “Patterns of survivor presentation for violence at health services” feels overly broad. It may be more accurate and helpful to name the specific categories within the theme itself (naming what are the patterns).

Theme 2b Women’s experience: As with Theme 2a (“Patterns”), the title of this theme (“Women’s experiences…”) does not clearly convey its core content. I recommend revising the theme title to more directly reflect the main finding or insight it represents, so that readers can immediately grasp its relevance and focus (e.g., what are the experiences of women?).

DISCUSSION

The stated aim of the scoping review is to identify opportunities for strengthening VAW health services during outbreaks in LMICs. However, this objective does not appear to be fully addressed in the current analysis, beyond a general call for more research. I recommend either revisiting the discussion to more explicitly identify and reflect on potential opportunities emerging from the findings, or reframing the study aim to better align with the scope and outcomes of the review.

The discussion presents findings at a general level, yet the LMIC contexts included in the review are highly diverse in terms of pandemic impact, existing health infrastructure, cultural norms, and how gender-based violence and VAW is understood and addressed. Grouping these contexts together without acknowledging this heterogeneity may oversimplify important contextual differences. I suggest addressing this point in the discussion or the limitations section, with reflection on how these variations may have influenced both the experiences reported and the generalizability of the findings.

The authors note that all included studies focused on experiences during the COVID-19 pandemic. It would be helpful to discuss why no studies were found on VAW service delivery during other outbreaks, especially given the “calls over the previous decade” (line 307) to ensure continuity of essential health services for women during such crises, and “longstanding consensus that life-saving services for survivors of violence need to be provided during outbreaks”. I am surprised to read that despite this consensus, there are no studies addressing this topic.

This might also be addressed in the synthesis section in the Methodology, but I am wondering about the inclusion of diverse study designs that capture both the perspectives of VAW service users (survivors) and service providers. While including both viewpoints is valuable, it would be important to explicitly acknowledge and discuss how these perspectives may differ or converge, even if they ultimately point to similar conclusions. Exploring these dynamics could add depth to the analysis and strengthen the interpretation of findings.

While this may fall slightly outside the scope of the study, it would be valuable to reflect on whose voices are represented in the included literature. Specifically, are the studies led by researchers based in LMICs, or primarily by teams from non-LMIC contexts? Considering this could offer important insight into how power dynamics and positionality might influence the interpretation and framing of findings.

MINOR COMMENTS

OVERALL

I recommend that authors take the time to review the manuscript, as I spotted multiple punctuation errors, and some grammar mistakes as well.

METHODOLOGY

Search strategy

Line 81 “between 2014 to May 2024”: Add the month for 2014 as well.

Lines 81-83: Specify that this timeframe also captures the COVID-19 outbreak (in addition to Ebola 2014-2016 and Zika 2015-2016)

Table 1 population(s) “LMICs”: Please specify how LMICs are defined or which classification was used to defined LMICs.

Abstract, line 112 and PRISMA figure: The number of included articles varies (N=32 in abstract and PRISMA, N=34 in text line 112). To be standardized.

RESULTS

Table 2: Authors identified some papers with an asterisk as “key papers”, but I cannot find information about what this does imply (key for what)? Please specify.

Table 2: I recommend adding more details about the design of studies instead of simply naming “qualitative” “mixed” etc. Sometimes “interviews” are specified for qualitative studies, sometimes, no details are provided. Adding more details would help readers better understand the scope and reach of each study. Please also make sure to review how the data is presented (e.g., capitalized or not).

Table 2: I highly recommend adding the number of participants (N) for each study to help readers better grasp the scope of the studies.

Table 2: It would be helpful to provide more information on the context of each study (e.g., aim of study, population, etc.)

Theme 1a. Disruptions

Line 28: “were also closed in non-humanitarian during”: A word seems to be missing.

Lines 34-36 “In Lebanon high rates […]”: This sentence is unclear (run-on) and grammatically incorrect – please address.

Lines 36-41: The sentences beginning with “In Kenya, Uganda […]” and “In Kenya, health workers […]” appear to convey overlapping information. I recommend revising for concision by combining or streamlining these sentences to avoid redundancy.

Lines 57-61, last paragraph of the Disruption theme: It is unclear how this paragraph fits within the theme of “disruption of services.” If the intention is to highlight services that continued or adapted successfully despite disruptions, I suggest reframing the paragraph to reflect that explicitly. It would also be helpful to briefly explain the factors that enabled these services to thrive in contrast to those that were disrupted.

Theme 1b. Adaptation

Line 68: Please define the concept of “gender digital divide”.

Line 119: The connection between the 2020 Beirut explosion and the actions of the GBV taskforce is unclear. Is the explosion mentioned to highlight the compounded challenges Lebanon faced, or did it directly influence the adaptations described? Clarifying this connection would help strengthen the coherence of the paragraph and its relevance to the theme of service adaptations.

Theme 1c. Continuation

Because paragraphs “In Brazil, primary […]” (line 147) and “Referral by health providers was […]” (line 154) both address the referral issue, I would advise combining both paragraphs into one. Another paragraph could be made for the point about the “perception that counselling was for ‘mad’ people” and for the non-existence of services regardless of COVID-19.

Theme 2b. Women’s experiences

2nd paragraph, line 227: I recommend answering the first sub-objective (“describe the prevalence”) by reporting the prevalence before reporting the type of help sought (family, friends, neighbours, etc.)

CONCLUSION

Line 364: Please add “COVID-19” to specify which pandemic.

Reviewer #3: The authors discuss several pandemics, but only COVID-19 is mentioned. If there is no data on other pandemics, it would be appropriate to limit the research to COVID-19.

Furthermore, the fact that the research covers several cases limits your results. It is preferable to limit your work to a single case so that it contributes more to science. The current analysis is superficial and omits many situations. Furthermore, I believe that the situations mentioned are not specific to low- and middle-income countries. In addition, I note that you are consulting documents that are not specific to violence against women.

6. PLOS authors have the option to publish the peer review history of their article (what does this mean?). If published, this will include your full peer review and any attached files.). If published, this will include your full peer review and any attached files.

**Do you want your identity to be public for this peer review?** For information about this choice, including consent withdrawal, please see our Privacy Policy..

Reviewer #1: **Yes:** Vena JosephVena Joseph

Reviewer #2: No

Reviewer #3: No

---

## [Decision Letter · Decision Letter 1]

18 Dec 2025

PGPH-D-25-00671R1

Health service responses and help-seeking for women experiencing violence during outbreaks in low- and middle-income settings: a scoping review

Dear Dr. Burns,

Thank you for submitting your manuscript to PLOS Global Public Health. After careful consideration, we feel that it has merit but does not fully meet PLOS Global Public Health’s publication criteria as it currently stands. Therefore, we invite you to submit a revised version of the manuscript that addresses the points raised during the review process.

We look forward to receiving your revised manuscript.

Kind regards,

Muriel Mac-Seing, PhD

Academic Editor

Journal Requirements:

Additional Editor Comments (if provided):

Dear Authors,

Thank you for submitting your revised manuscript.

Please find the reviewers’ feedback indicating a "minor revision" to further refine your manuscript. A few items are left to be clarified.

Best,

Muriel Mac-Seing

Reviewers' comments:

Reviewer's Responses to Questions

**Comments to the Author**

1. If the authors have adequately addressed your comments raised in a previous round of review and you feel that this manuscript is now acceptable for publication, you may indicate that here to bypass the “Comments to the Author” section, enter your conflict of interest statement in the “Confidential to Editor” section, and submit your "Accept" recommendation.

Reviewer #2: (No Response)

Reviewer #3: All comments have been addressed

2. Does this manuscript meet PLOS Global Public Health’s publication criteria? Is the manuscript technically sound, and do the data support the conclusions? The manuscript must describe methodologically and ethically rigorous research with conclusions that are appropriately drawn based on the data presented.? Is the manuscript technically sound, and do the data support the conclusions? The manuscript must describe methodologically and ethically rigorous research with conclusions that are appropriately drawn based on the data presented.

Reviewer #2: Yes

Reviewer #3: Yes

3. Has the statistical analysis been performed appropriately and rigorously?

Reviewer #2: N/A

Reviewer #3: N/A

4. Have the authors made all data underlying the findings in their manuscript fully available (please refer to the Data Availability Statement at the start of the manuscript PDF file)?

The PLOS Data policy requires authors to make all data underlying the findings described in their manuscript fully available without restriction, with rare exception. The data should be provided as part of the manuscript or its supporting information, or deposited to a public repository. For example, in addition to summary statistics, the data points behind means, medians and variance measures should be available. If there are restrictions on publicly sharing data—e.g. participant privacy or use of data from a third party—those must be specified.requires authors to make all data underlying the findings described in their manuscript fully available without restriction, with rare exception. The data should be provided as part of the manuscript or its supporting information, or deposited to a public repository. For example, in addition to summary statistics, the data points behind means, medians and variance measures should be available. If there are restrictions on publicly sharing data—e.g. participant privacy or use of data from a third party—those must be specified.

Reviewer #2: Yes

Reviewer #3: Yes

5. Is the manuscript presented in an intelligible fashion and written in standard English?

Reviewer #2: Yes

Reviewer #3: Yes

6. Review Comments to the Author

Reviewer #2: I would like to thank the authors for responding to my previous comments and for providing clear and thoughtful answers. Below, I offer additional remarks that I believe could further strengthen the manuscript. Most of these are minor comments and suggestions. The major issues concern the alignment between the methodology described for data analysis and the formulation of the themes and resulting findings. At present, the methodology does not fully correspond to the results, which constitutes a relatively significant methodological issue. I strongly encourage the authors to address the major recommendations provided.

1-Minor - Various locations - “women’s help-seeking for violence” seems a bit clumsy as it can sound like women are seeking help to commit violence rather than seeking help because they are experiencing violence. We recommend rephrasing (e.g., help-seeking related to violence, help-seeking behaviors in response to violence, help-seeking among women experiencing violence, etc.)

2-Minor - Various locations - Please proofread this newer version of the manuscript, there are many punctuation mistakes, for example:

- A space is missing in “weredeprioritised” (line 48)

-commas are missing after “e.g.” (e.g., line 65, Table 1)

-There often appear to be two spaces in a row. Please correct this (e.g., line 68 and line 71 after ‘VAW’)

- Make sure you use em dash rather than hyphen for parenthetical thoughts (e.g., line 69 “such as Ebola […]”)

-Avoid using two punctuation marks following one another (e.g., line 74 ?.)

-I think the capitals are unnecessary in “Eighteen interviews” and “Epidemiological Surveillance” (Table 2)

-Most objective/aim in Table 2 ends with a period, except Decjer et al., 2022. Please standardize.

-Some Study periods end with a period (e.g., Vahedi et al., 2023; Yirgu et al., 2023), others don’t. Please standardize

-Line 89 in the results: remove parentheses at the end of the sentence

-Commas are missing, for example in the “Similarly in Bangladesh the United […]” sentence (Results line 144) or “In another study in Kenya data was […]” sentence (Results line 178)

-“s” is missing from “health workerS strike” (line 180 Results)

-comma missing after “Uganda” (line 281)

-commas missing in sentence on lines 368-371 (Mpox limitation) and it runs two independent clauses together without proper separation. I recommend diving the sentence in two.

Other mistakes were also spotted so a thorough read would help correct them.

3-Minor - Abstract - This sentence is confusing to read “Understand how health services for violence against women (VAW) were affected in low- and middle-income (LMIC) settings during recent outbreaks, and women’s help-seeking for violence.” Please consider rewriting it (e.g., use the phrasing used in the background section linee 62-64)

4-Minor - Background line 38 - Please define VAW, ideally in relation to IPV and GBV (see comment #8). It will help understand its connection with IPV, GBV and SGBV.

5-Minor - Background line 54 - Please specify if “80% of countries” concerns all countries or only LMIC

6-Minor- Background line 58 - I am not sure “policy briefs” (and advocacy document) can be described as “normative efforts”, which would be norms, guidelines. Please refine the wording.

7-Minor - Methods lines 92-94 - Please add references to support this statement “[…], as these outbreaks met these criteria and have been shown in prior research to exacerbate the risks of VAW and barriers to accessing care.”

8-Minor - Methods lines 98-100 - GBV is defined, but it’s not specified whether it’s included or not in VAW. Please specify. At this point, it’s unclear to me how the concepts of GBV, VAW and IPV are related especially in this study (e.g., are VAW and IPV subtypes of GBV? Then, wouldn’t the inclusion of GBV already include VAW? But if this scoping review is focused on VAW, then it would exclude other types of GBV?)

9-Minor - Methods Table 1 - In the population(s) row, are both cis and trans women included (would you have excluded trans women to focus on cis women)? Please specify.

10-Minor Methods Table 1 Please specify what “members of affected communities” mean. Affected by VAW? GBV? SGBV? Affected by the pandemic?

11-Minor - Methods Table 1 (Limitation section, line 365) -Focusing on concept 1 (health service search terms) may have limited the findings about women’s help-seeking patterns and behaviors (2nd objective and research question). Ideally, a second specific search strategy would have been used for the second objective and research question (excluding the 1st concept). Specifically, in the “intervention(s)” section of Table 1, only studies looking at access to, delivery of health services interventions, health outreach/awareness programs are included and only studies describing health services (“outcome(s)” row) are included. Women may have sought help outside of these formal programs/interventions, so limiting the search to this concept is a limitation to the understanding of help-seeking patterns and behaviors. Please address it in the limitation section.

12-Minor - Methods Table 1 - Would reviews reporting primary results be included? If not, please add “reviews” in the “exclusion” column.

13-Minor - Figure 1 (PRISMA) - PRISMA figure incomplete. Many numbers are missing (blank) from the PRISMA figure, including the number of articles excluded based on titles and abstracts screening, the articles sought for retrieval, article not retrieved, article assessed in full text for eligibility. Please complete the PRISMA figure.

14-Minor - Synthesis line 121 - “supply” and “demand” are not themes, but rather categories. Please address the vocabulary used.

15-Minor - Synthesis line 123 - Themes cannot really “emerge”; rather, they are constructed/produced by authors/researchers. See Ahmed et al., 2025: https://www.sciencedirect.com/science/article/pii/S2949916X25000222 Please adapt the language accordingly.

16-Minor - Synthesis line 123 - “This analysis was initially based on the themes emerging from the five papers listed as ‘key papers’ “: I am unsure what authors refer to when writing “themes emerging”: themes generated by the authors of the included original papers? Of themes generated by the authors of this scoping review?

Additionally, “five papers listed as key papers” requires additional information: please explain what those key papers are (e.g., refer to the Results section to indicate that these papers will be presented later)

17-Major - Synthesis line 131 - Line 122 says “themes” (categories) were inductively divided into sub-themes. However, the aim of the study was to “to identify disruptions, adaptations” (lines 69-70)… which end up being the “themes.” This indicates that the analysis was not inductive but entirely deductive (the categories are the same as the predetermined goal). This would be acceptable if presented as a deductive content or thematic analysis rather than an inductive thematic analysis. Please address this as this is a major methodological error.

18-Minor - Synthesis line 127 - Please explain briefly the Thomas & Harden approach and how it was used in this work.

19-Major - Synthesis 134-135 - Repeating myself from the first round of comment, but this is key: “ii. women’s experiences seeking help with informal sources” is not a theme, it is a category. I highly recommend adapting the analysis section to represent seemed to have been done (content analysis rather than thematic analysis) OR working on the categories so they adequately represent themes (a theme, here, would clearly express what the “women’s experiences” are to represent the essence of those experiences, instead of having a broad term like “women’s experiences” that could mean literally anything [a category]).

This was a comment that I previously made in the first round, but is crucial, as what is presented as “thematic analysis” in this manuscript is not.

20-Major - Synthesis line 137 (and throughout the Results section) - As I now understand that authors analyzed secondary constructs only, I would strongly advise not using primary construct (direct participants’ quotes) to “illustrate authors findings”. If authors want to illustrate their findings (stemming from secondary constructs), I recommend using quotes from secondary constructs only, as this is what was analyzed. I think another reviewer recommended this as well.

21-Minor - Results line 208- Thank you for this addition. Would it be possible to add the total number of studies where 1st author are from LMIC institutions?

22-Minor - Results lines 215-217- I suggest adding the Arksey and O’Malley reference to support this claim.

23-Minor - Table 2, Study aim column - I suggest removing the “To” in front of all listed objectives/aim to lighten the table just a little bit.

24-Minor - Table 2 - The articles aren’t presented in alphabetical or chronological order in Table 2. I would advise using the alphabetical order.

25-Minor - Throughout Results section - Please add the reference from which the quotes are from (that should be from secondary constructs only) at the end of the quote.

26-Minor - Throughout Results section - Thank you for including for contextual information for the findings. When/if possible, please be more specific about the context of the findings as simply stating the country might not be enough (contexts can vary a lot inside the countries). For example, line 34 (young women participant, Kenya), should read (city of Kisumu, Kenya). Same applies with all quotes (e.g., “Before coronavirus I used to tell […]”; “There are families […]”) This would help better contextualize findings.

Although I recommend removing quotes from primary constructs, this applies to quotes from secondary constructs if some were to be added.

27-Minor - Results line 60 - Although well known, please define USAID acronym.

28-Minor - Results line 63 - The sentence “A USAID funded project […] average of 642%” is unclear (increased compared to what? A % cannot be a monthly count). Please rephrase (e.g., 642% average monthly increase in the identification, management, and response to GBV survivors, compared to pre-intervention levels)

29-Minor - Results 2a (line 172)- I feel like the 2a “theme” phrasing is a bit clunky. Here are some suggestions: Evolving patterns of help-seeking; Variability in survivors’ engagement with services; Outbreak-driven shifts in survivors’ access to services

30-Minor - Discussion lines 298-299 - Please cite evidence showing that high-income countries got public/media attention earlier than LMICs for the information in parenthesis (“initially in high income[…]”).

31-Minor Limitation section- You might want to add that the text screening process was done with a single reviewer (rather than two independent reviewers), which may have introduced a bias.

32-Minor - Discussion line 333 - Last sentence of paragraph doesn’t seem idiomatic (“linked with a review”) and is unclear to me. I recommend rephrasing.

33-Minor - Conclusions line 382 - Please specify what the ‘good practices’ refer to (e.g., VAW services good practices?)

Reviewer #3: This second version of the article seems to me to be an improvement. We felt that the choice of low- and middle-income sites was not justified and constituted a bias in the study. Improvements have been made in this regard. I note an effort to improve the work. However, I remain convinced that the scientific contribution of this work would have been significant if the authors had opted for a comparative study between low-, medium-, and high-income situations. I believe that highlighting the different specificities of each country would have been of definite scientific interest and would be of paramount importance in terms of programming. In view of the efforts made, I think this article should be accepted with minor revisions.

7. PLOS authors have the option to publish the peer review history of their article (what does this mean?). If published, this will include your full peer review and any attached files.). If published, this will include your full peer review and any attached files.

**Do you want your identity to be public for this peer review?** For information about this choice, including consent withdrawal, please see our Privacy Policy..

Reviewer #2: No

Reviewer #3: No

 Figure Resubmissions:

---

## [Editor Report · Decision Letter 2]

31 Mar 2026

Health service responses and help-seeking for women experiencing violence during outbreaks in low- and middle-income settings: a scoping review

PGPH-D-25-00671R2

Dear Dr Burns,

We are pleased to inform you that your manuscript 'Health service responses and help-seeking for women experiencing violence during outbreaks in low- and middle-income settings: a scoping review' has been provisionally accepted for publication in PLOS Global Public Health. **Please conduct a final proofread for typos, including correcting the misspelling of "restrictions" on page 30, line 312, and removing the word "social" from "social or physical distancing" on lines 28 and 313 to ensure consistency with "physical distancing" throughout the manuscript.**

Best regards,

Muriel Mac-Seing, PhD

Academic Editor